# Bone Marrow Aspirate Concentrate: Its Uses in Osteoarthritis

**DOI:** 10.3390/ijms21093224

**Published:** 2020-05-02

**Authors:** Gi Beom Kim, Min-Soo Seo, Wook Tae Park, Gun Woo Lee

**Affiliations:** 1Department of Orthopedic Surgery, Yeungnam University College of Medicine, Yeungnam University Medical Center, 170 Hyonchung-ro, Namgu, Daegu 42415, Korea; donggamgb@hanmail.net (G.B.K.); gom2ya@gmail.com (W.T.P.); 2Laboratory Animal Center, Daegu-Gyeongbuk Medical Innovation Foundation (DGMIF), Daegu 41061, Korea; msseo@dgmif.re.kr

**Keywords:** bone marrow, bone marrow aspirate concentrate, mesenchymal stem cells, osteoarthritis, cartilage, regeneration

## Abstract

Human bone marrow (BM) is a kind of source of mesenchymal stem cells (MSCs) as well as growth factors and cytokines that may aid anti-inflammation and regeneration for various tissues, including cartilage and bone. However, since MSCs in BM usually occupy only a small fraction (0.001%) of nucleated cells, bone marrow aspirate concentrate (BMAC) for cartilage pathologies, such as cartilage degeneration, defect, and osteoarthritis, have gained considerable recognition in the last few years due to its potential benefits including disease modifying and regenerative capacity. Although further research with well-designed, randomized, controlled clinical trials is needed to elucidate the exact mechanism of BMAC, this may have the most noteworthy effect in patients with osteoarthritis. The purpose of this article is to review the general characteristics of BMAC, including its constituent, action mechanisms, and related issues. Moreover, this article aims to summarize the clinical outcomes of BMAC reported to date.

## 1. Introduction

Articular cartilage has been well-known for low spontaneous healing potential since it may lack vessels and undifferentiated cells, and the presence of specialized cells with low mitotic activity [1,2]. Therefore, once damaged, it may eventually progress to osteoarthritis (OA). OA is a destructive joint disease, causing degeneration of cartilage, osteophyte formation, and changes in periarticular bone, resulting in disability [3,4]. To our knowledge, there has been no approved, established treatment that can reverse the progression of OA or destruction of articular cartilage [5]. Although several studies have been conducted on disease modifying strategies of OA at the molecular level that block inflammatory pathways and enhance cartilage protective function [6,7], there have still been limitations in establishing the optimal treatment options. Moreover, from microfracture to osteochondral autologous transplantation, various surgical techniques for cartilage regeneration have been introduced [8,9,10]. However, the current therapies are still palliative, and there has been no optimal regenerative method for OA with cartilage degeneration.

Recently, bone marrow aspirate concentrate (BMAC) has emerged as a possible alternative for regenerative medicine. It has been spotlighted as a promising biologic tool because of a rich source of pluripotent mesenchymal stem cells (MSCs) and growth factors [11,12], and currently approved by the United States Food and Drug Administration (FDA). Accordingly, considering both the anti-inflammatory and regenerative effect, BMAC may be an attractive tool for cartilage regeneration in OA. Although several studies on BMAC have been introduced, there has been paucity in the systemically organized review on its overall contents. Therefore, the purpose of this article is to review the general characteristics of BMAC, including its constituent and action mechanisms. Moreover, this article aims to summarize the clinical outcomes of BMAC for OA with cartilage degeneration reported to date.

## 2. Constitution

### 2.1. Cellular Contents in Bone Marrow Aspirate (BMA) and BMAC

Bone marrow (BM) primarily produces the red blood cells during hematopoiesis. It mainly consists of hematopoietic cells, adipose tissue (adipocyte), and supportive stromal cells [13]. The organized stroma of the BM promotes the proliferation and differentiation of the hematopoietic cells, and contains supporting cells such as fibroblasts, macrophages, adipocytes, osteoblasts, osteoclasts, and endothelial cells [14]. Particularly, the BM also contains mesenchymal stem cells (MSCs) or marrow stromal cells. Numerous studies have documented that MSC in BM itself is insufficient not to have a therapeutic property for chondral regeneration. For cartilage or chondral regeneration, several methods have attempted delivering the reparable agents, such as MSC, to the defect area. Microfracture or multiple drilling techniques have been widely performed, relying on migration of chondrogenic (precursor) cells like MSC from subchondral bone marrow to the target area of articular surface. However, this method results in fibrocartilage rather than hyaline cartilage, due to an insufficient amount of chondrogenic cells, MSCs, and its poor concentration. Generally, since fibrocartilage has low mechanical strength and a small number of cartilage cells, orthopaedic surgeons have tried to discover advanced methods for replacing the chondral defect with hyaline cartilage, rather than fibrous cartilage [15,16].

Recently, several solutions have been invented to overcome the insufficient amount of MSCs (e.g., chondral cells). Among them, BMAC can produce higher concentration of chondrogenic cells, MSCs, and other affirmative stromal cells, in comparison with BM itself [17]. In addition to MSCs, BMAC contains growth factors more than BM itself, which has been proven to be equal or even superior to other conventional cartilage restoration techniques [18]. Moreover, BMAC also have numerous bioactive molecules and cell types including lymphocytes, neutrophils, monocytes, and platelets in various stages of differentiation [19]. In cytological analysis, BMAC contains an increased number of platelets and white blood cells [20]. Johnson et al. [21] demonstrated that a fourfold increase in platelets, total nucleated cells, and CD34+ cells in BMAC compared to BMA was reported.

### 2.2. Mesenchymal Stem Cells (MSCs)

MSCs are multipotent progenitor cells that can be obtained from bone marrow, adipose tissue, synovium, articular cartilage, and skeletal muscles [22,23]. As only about 0.001% of all nucleated cells in bone marrow (BM) are MSCs [24]. MSCs present self-renewal and differentiation potential into cells of the mesodermal lineage such as cartilage, bone, fat, muscle, meniscus, and ligaments [25,26]. The Mesenchymal and Tissue Stem Cell Committee of the International Society for Cellular Therapy defined the MSCs as the following criteria: (1) be plastic-adherent in standard culture conditions; (2) express CD105, CD73, and CD90 at their surface; (3) be lack expression of CD45, CD34, CD14 (or CD11b), CD79α (or CD19), and HLA-DR; and (4) differentiate into osteoblasts, adipocytes, and chondroblasts in vitro (Table 1) [27]. If these criteria are not met, the term “MSCs” cannot be used.

MSC can be differentiated into several different cell types, including osteoblasts, adipocytes, chondroblasts, or even neurogenic cells, under specific environments for differentiation. In particular, chondrogenic property from BM can be driven by using specific insulin, transferrin, selenium, transforming growth factor beta (TGF-*β*), and others. Among the specific materials, the TGF-*β* family (three isoforms) has been recognized as a critical factor for cartilage development and genesis [28]. In addition, recent studies have shown that TGF-*β* family is also an important source for composing type II collagen that is composed of extracelluar matrix in cartilage [28,29]. More detailed information (growth factors, cytokines, and other factors) regarding cartilage regeneration among constituents in BM is described below.

### 2.3. Growth Factors and Cytokines

BMAC also serves as a rich source of factors that can influence the healing responses by decrease in cell apoptosis and inflammation, and by activation of cell proliferation, differentiation, and angiogenesis via paracrine and autocrine pathways [11,12]. Numerous factors involved in these trophic processes include the platelet-derived growth factor (PDGF), TGF-*β*, vascular endothelial growth factor (VEGF), fibroblast growth factor (FGF), insulin-like growth factor I (IGF-I), granulocyte-macrophage colony-stimulating factor (GMCSF), bone morphogenetic protein (BMP-2 and 7), and interleukins (IL-1β, 6, 8) [20,30,31]. These bioactive factors are assigned to have anabolic and anti-inflammatory effects, resulting in positive effects on cartilage repair and treatment of OA. Specifically, PDGF can be a specific role in cartilage regeneration and maintaining homeostasis, via MSC proliferation and inhibition of IL-1*β* -induced chondrocyte apoptosis/inflammation pathway [32], while TGF-*β* family (three isoforms) has a role of stimulating chondrogenesis, inhibiting inflammation, and enhancing cartilage healing or regeneration [33]. Moreover, they may have the potential to express their specific activities via inter-molecular action and subsequently to promote MSC associated tissue healing.

## 3. Pathophysiology and Issues of Osteoarthritis

OA is a degenerative joint disease characterized by loss of cartilage, osteophyte formation, and periarticular bone change, resulting in disability [3,4]. Unfortunately, up to date, there has been no approved treatment that can reverse the progression of OA or destruction of articular cartilage. Recent advancement in cell-based treatment offers a new era for OA management. In order to establish the disease modifying (or managing) strategies of OA, it is necessary to consider the biomolecular features in OA circumstance and their relationship between MSC (specific materials) and OA.

Molecular and cellular mechanism in development and progression of OA have to be aware for establishing therapeutic plan with MSC [34]. Innate immune cells like natural killer cell or macrophages can play an important role in an early inflammatory phase [35]. TNF-α and IL-1β also have a function to shift tissue homeostasis towards catabolism by degradation, resulting in cartilage resorption [36,37]. In addition, mechanisms such as increased proinflammatory cytokines such as IL-1 or TNF-α, decreased growth factors such as TGF-β, activated matrix metalloproteinase, and ultimate chondrocyte senescence can be observed at the molecular level (Figure 1) [38,39,40].

## 4. Mechanisms of BMAC for Osteoarthritis

The action mechanism of BMAC is not yet fully understood [41]. In order to analyze the exact mechanism of BMAC, an understanding of the presence of MSCs in BMAC must be preceded. Most of all, the MSCs within BMAC will potentially provide a direct cell source for tissue repair. Additionally, the nucleated cells may have a paracrine effect by delivering various growth factors and cytokine into the lesion site to promote tissue healing and immunomodulation [41,42,43]. After density gradient centrifugation, the harvested cells may be concentrated six to seven times, so that cellular content in BMAC can explode several growth factors, such as PDGF, TGF-β, and VEGF [11]. These growth factors are within the α-granules of platelets and are secreted by MSCs [44], which have high chondrogenic potential [12,45]. PDGF, TGF-*β*, and other factors such as IGF-I also serve as chemoattractant.

MSCs also have an immunosuppressive effect by adjusting the activation of natural killer cells, dendritic cells, macrophages, and T and B lymphocytes [46,47,48]. Thus, MSCs have advantageous anti-inflammatory and antifibrotic actions to maximize their therapeutic effects in lesion site [49,50,51].

## 5. Issues on Harvest and Processing of BMAC

Since the regenerative capacity of BMAC has been closely linked to the number of MSCs present in the graft at the time of its clinical application [52], the methodology of the harvest or how to obtain a larger population of MSCs would be an important issue. Although MSC counts in BMAC may be affected by the harvest technique [53], there is still no consensus on the aspiration method optimizing cellular yield. Hernigou et al. [53] reported that higher concentrations of progenitor cells could be obtained with a smaller volume syringe (10 mL) with multiple site harvesting. On the contrary, a recent study found no significant difference in final cell concentration between single- and multiple-site harvesting techniques [52]. Additionally, the single-site technique was significantly less painful not only at the time of procedure but also after 24 h.

What amount or concentration of BMAC is the most efficient for chondral regeneration has not been determined as well. Yandow et al. [54] demonstrated that, in children, up to 5 mL bone marrow from the iliac crest yields a proportional increase in osteoblastic progenitor cells per aspirate, and increasing the aspiration volume beyond 5 mL primarily results in a hemodilution and loss of biological efficacy rather than further increased harvest of osteoblastic material. Similarly, Muschler et al. also studied that the effect of aspiration volume by comparing 1 mL, 2 mL, and 4 mL BM aspirate sample, and concluded that, although the total number of MSCs increased with greater aspirated volumes, so did the quantity of diluting peripheral blood [55]. Another study also demonstrated that an aspiration of 10% to 20% of the syringe volume was ideal [53]. Based on the reports, in most studies, the amount of BMAC extracted was 60 mL [56,57,58], but Kim et al. [59] reported extraction of 120 mL of BM. The harvested BM can be concentrated in one of many FDA approved devices (Figure 2). This process concentrates the buffy coat containing mononuclear cells and increases the number of MSCs relative to baseline [60]. Some studies [57,58,59] used a SmartPreP2 Bone Marrow Procedure Pack BMAC2 kits (Harvest Technology, Plymouth, Massachusetts, USA) for centrifugation, and then was mixed using batroxobin enzyme (Plateltex Act, Plateltex SRO, Bratislava, Slovakia). Others [56,61,62] processed their extracted BM using the MarrowStim Concentration Kit (Biomet, Warsaw, IN), obtaining 3–4 mL of BMAC.

## 6. Application Modalities

The application modalities of BMAC such as intra-articular injection or surgical implantation with bio-scaffold may also be an issue to be considered in the clinical settings. Intra-articular injection is relatively simple, easy to apply, and does not require hospitalization for the procedure [63,64]. However, accurate delivery to the lesion site can be difficult [19]. On the contrary, surgical implantation enables precise delivery to the lesion site, and acquires stability when combined with a bio-scaffold or membrane. The scaffold or membrane can reduce chondrocyte loss, maintain uniform cellular distribution, and ultimately enhance chondrogenesis. Their materials mainly used in studies are hyaluronic acid, collagen derivatives, agarose, fibrin glue, and chitosan [15,57,58,62]. Nevertheless, surgical implantation usually requires hospitalization as a more invasive method. Therefore, the optimal strategy has not yet been identified. Occasionally, intralesional injection may be an alternative option. Particularly, since degenerative cartilage deterioration secondary to osteonecrosis or avascular necrosis has subchondral bone pathology, implantation into the necrotic zone may be advantageous by reconstructing a micro-environment of osteoblast differentiation and endothelial cell proliferation [65,66,67].

## 7. Review of Clinical Studies with BMAC

Several BMAC studies on focal cartilage defects have reported favorable outcomes (Table 2) [15,56,57,58,61,62,68]. Gobbi et al. [57,58] reported complete coverage of cartilage lesions with hyaline-like features in 80% of patients. They confirmed with normal to nearly normal tissues on histologic findings performed concurrently with second-look arthroscopy. Recently, they reported excellent long-term clinical outcomes of hyaluronic acid-based scaffold embedded with BMAC in a full-thickness cartilage defect [68]. Gigante et al. [61] reported the outcomes of BMAC combined with microfracture. They showed good defect filling with normal tissue signal with no signs of bone marrow edema. Skowroński et al. [62] reported favorable clinical outcomes of BMAC with collagen membranes in large chondral lesions. However, Enea et al. [56] found bone marrow edema and subchondral irregularities in all patients, even though histology showed hyaline-like repair tissue.

Few studies have evaluated the outcomes of BMAC injection in patients with knee OA (Table 3) [59,69,70]. Hauser et al. [70] reported the clinical outcomes of intra-articular injection of unfractionated BM combined with hyperosmotic dextrose in patients with OA. They showed that complete relief or functional improvement at least six weeks follow-up in 5 of 7 patients. Centeno et al. [69] used autologous BMAC for intra-articular injection with or without adipose grafts. Their study reported that patients with Kellgren–Lawrence (K–L) grades I or II showed significantly better improvement in clinical outcomes than K–L grades III–IV. In a systematic review, Chahla et al. [71] suggested that BMAC treatment would be a safe procedure with good results reported. Conversely, a recent study with prospective, single-blind, placebo-controlled pilot study (BMAC vs. saline) was performed in patients with bilateral OA [72]. There was no significant difference in pain relief and function between the both sides. However, it is difficult to directly compare the differences because of heterogeneity across studies including the study design, sources and doses of cells, and the administration of adjuvant therapy. Further studies with prospective, randomized, double-blinded fashion, and larger sample size are required to gain a better understating of BMAC treatment. To date, most of the BMAC-related studies have focused on functional improvement rather than quality of regeneration. To assess the regenerative effects of BMAC, further radiologic or histologic analyses of remodeled cartilage are necessary.

The authors have also reported the short-term clinical and functional outcomes of intra-articular BMAC injection mixed with adipose tissue matrix as a scaffold in a case series of 41 patients (75 knees) with knee OA (K–L grades I–IV) [56]. At 12-month follow-up, BMAC injection significantly improved knee pain and function. Particularly, a significant relationship was found between superior outcomes and lower K–L grade at follow-up. However, to compensate for the limitation of short-term follow-up, longer-term outcomes of intra-articular BMAC injection is necessary.

## 8. Adverse Events

Most of the reported complications for BMAC were nonspecific and self-limited, with the symptoms including pain, swelling, skin rash or itching, and aspirate site problems. Few severe complications have been reported and all the events generally resolved without any intervention. Centeno et al. [63] reported the frequency of adverse events after the procedure to be 6% for BMAC alone. Knee joint pain and swelling were the most common adverse events.

## 9. Advanced Technologies

### 9.1. Nanotechnologies

Nanotechnology or nanoengineering refers to the science of manipulating nanometer-sized materials (1–100 nm), and involves a number of nanomaterials used in various fields of regenerative medicine, including tissue engineering, cytotherapy, and drug or gene delivery [73]. At the nano-scaled level, materials have different unique physical and chemical properties compared to those at larger scales [74]. By integrating the nanomaterials, it facilitates developing the novel scaffolds that stimulate the extracellular matrix (ECM) environment around the native cartilage to promote the cell-scaffold interaction and improve the functionality of the tissue engineered constructs [75]. The surrounding environment consists of nano-scaled particles that provide multiple biological signals, which ultimately affect cell behavior, resulting in cell shape, cell skeleton, and focal adhesion [75,76]. This technique can be applied to the degenerative OA joints as well as focal cartilage defects.

Nano-engineering combined with cell-based biology is the key to regenerative medicine. Up to date, the most commonly used scaffolds were natural materials such as HA or collagen-based materials [77,78]. Natural materials are derived from human or animal sources and consist of extracellular components. They include agarose, chitosan, alginate, silk protein, and fibrin glue [79,80,81,82]. Although they have shown promising outcomes, there have been disadvantages of mechanical properties, immunogenicity, and contamination [83]. Additionally, the outcomes of clinical studies related to the application of these materials have been still unsatisfactory [84].

Therefore, synthetic compounds are emerging as an alternative to regenerative medicine. Various types of synthetic scaffolds including poly ethylene glycol, poly polylactide acid and derivatives, polyurethane, and poly vinyl alcohol have also been utilized [85]. Compared to natural materials, they provide excellent controlled mechanical properties [86]. The robust scaffolds are generally more suitable for load-bearing cartilage engineering. By grafting nanotechnology, these novel scaffolds can exhibit superior properties of biocompatibility, controlled porosity and permeability, mechanical suitability for the target tissue, and support for cell attachment and proliferation [87,88]. Nano-structured surfaces produce nano-topography that can accelerate cell adhesion and proliferation than untreated surfaces [89]. Furthermore, nanometer surfaces with roughness may improve endothelial cell functions as compared to smooth polymer surfaces [90].

### 9.2. Other Smart Materials

To complement the rigidity for application of existing solid scaffold materials, synthetic or naturally derived hydrogels have gained popularity as a smart scaffold material due to their ability to transport oxygen through diffusion and to integrate into the ECM [91]. Because of their innate hydrated structure, hydrogels are typically biodegradable in physiologic environments, have mechanical and structural properties similar to the ECM, and can be delivered in a minimally invasive manner [85]. Either alone or combined with other type of cells, they have been utilized as in vivo implants to promote the regeneration of local tissue, ranging from peripheral nerve to skin substitute [92]. Although several studies have been conducted to improve the functionality of hydrogel through incorporation with HA, collagen, and fibrin [93,94,95], further researches is needed to improve the capability to preserve by supplementing their degradation properties.

Recently, thermo-reversible HA hydrogels have been introduced as an attractive candidate of smart materials. Through cross-links of modified HA and thermo-responsive poly, it has a gelling temperature for which the assembly of these modified HA chains can be altered and have good biocompatibility [96]. Additionally, it has been reported that they be can removed through renal excretion and be inserted in the site of injury through simple intra-articular injection [85].

Patient-specific three-dimensional (3D) bioprinting has gained due attention in tissue engineering for its ability to spatially control the location of cells, biomaterials, and other biological molecules [97]. This technology enables bioproduction of target tissues or organs specific 3D structures that combined with computer-aided design or manufacturing using the patient’s own medical images [98]. Computer-aided 3D-constructs are able to not only be more suitable for shape and size, but also enhance the cell viability and proliferation.

## 10. Concluding Remarks and Perspectives

In conclusion, MSCs within BMAC have the self-renewal capacity, can undertake clonal expansion, and differentiate into various mesodermal tissues. MSCs are also a rich source of several growth factors and cytokines, which have a paracrine and immunomodulatory effect. For this reason, BMAC has emerged as a promising biologic tool for regenerative medicine. Studies published to date have reported relatively favorable outcomes, but most of them have focused on clinical improvement rather than quality of regeneration. Moreover, heterogeneity between the studies may not allow for direct comparison.

Accordingly, there is a need for well-designed, randomized, controlled trials with large sample sizes to further evaluate the therapeutic action of BMAC for knee pathologies. Such studies can provide a better understanding of safety, aspiration amount, and the need for bio-scaffold, to ensure consistent and reproducible results of BMAC treatment. Additionally, the development of novel bio-scaffolds through grafting with nanotechnology or computer-aided technology can be a good opportunity to expand the clinical application of BMAC.

## Figures and Tables

**Figure 1 ijms-21-03224-f001:**
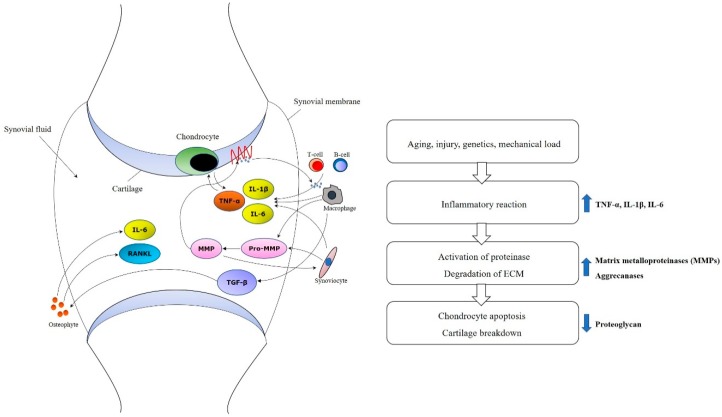
Molecular mechanisms of osteoarthritis. Increased proinflammatory cytokines such as TNF-α, IL-1β and IL-6, activated matrix metalloproteinases (MMPs), and decreased growth factors such as TGF-β and ultimate chondrocyte senescence can be observed at the molecular level.

**Figure 2 ijms-21-03224-f002:**
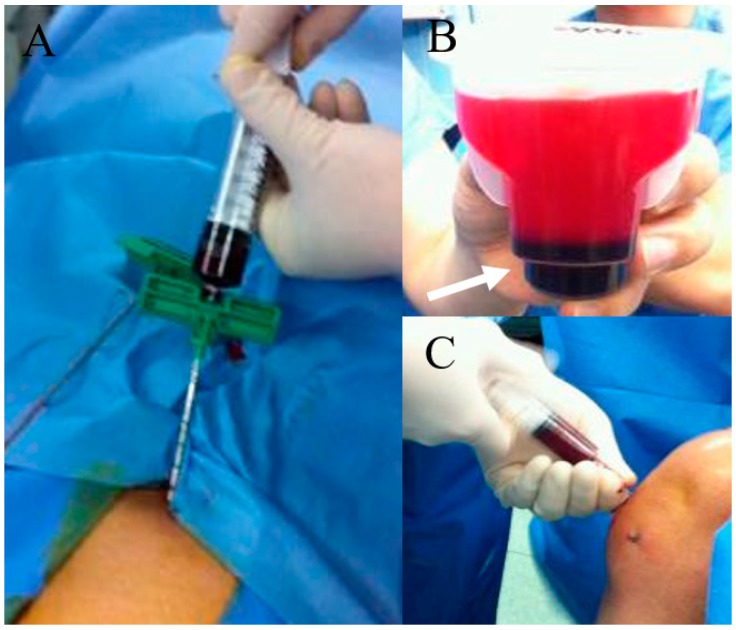
BMAC preparation and knee joint injection. (**A**) bone marrow aspiration at anterior iliac rim; (**B**) After centrifugation and some procedure, dark-colored BMAC (white arrow) was obtained; (**C**) BMAC injection to the knee joint with osteoarthritis.

**Table 1 ijms-21-03224-t001:** Standard criteria of mesenchymal stem cells defined by the Mesenchymal and Tissue Stem Cell Committee of the International Society for Cellular Therapy [27].

Characteristics	Positive Markers	Negative Markers
Plastic adherent in vitro	CD44	
Ability to form colony forming fibroblast	CD73	CD34
Ability to differentiate into mesodermal lineages (osteoblasts, adipocytes, chondroblasts, and tenocytes)	CD90 CD105	CD45 CD133
Promotion of hematopoiesis	CD166	
Self-renewal potential	HLA-ABC	

CD, cluster of differentiation; HLA, human leukocyte antigen.

**Table 2 ijms-21-03224-t002:** Details of the clinical trials of BMAC focusing on cartilage defects.

Authors	Publications/Year	Study Design/No. of Patients	Mean Age (Year)	Mean Follow-Up Period (Months)	Defect/Location	Treatment	Additional Factors	Harvest Volume/Kit Used	Outcomes	Complications
Buda et al. [15]	J Bone Joint Surg Am/2010	Case series/20	N/A (15‒50)	24.0	ICRS grade III–IV lesion/MFC and LFC	BMAC + HA membrane	Platelet gel (platelet rich fibrin)	60 mL/SmartPrep System	Significant clinical improvement; Subchondral bone & cartilage regeneration on MRI/histology	N/A
Gigante et al. [61]	Arthrosc Tech/2012	A case report	37.0	24.0	3.0 cm^2^ sized ICRS grade IV lesion/MFC	BMAC + fibrin glue	Microfx.	60 mL/MarrowStim Concentration Kit	Asymptomatic; MRI at 12 months showed good defect filling with normal signal	N/A
Skowroński et al. [62]	Orthop Traumatol Rehabil/2013	Retrospective comparative study/46	26.0	60.0	>4cm^2^ width & >6mm deep/MFC	BMAC (21) vs. Peripheral blood MSCs (25)	Autologous spongy bone graft, collagen membrane	27 mL/MarrowStim Concentration Kit	Clinical improvement in both groups; Peripheral blood MSCs group had superior results; Confirmed cartilage integration on MRI	N/A
Gobbi et al. [58]	Am J Sports Med/2014	Case series/25	46.5	41.3	Mean 8.3 cm^2^ sized ICRS grade IV lesion/MFC or patellar or trochlea	BMAC	Collagen membrane + fibrin glue	60 mL/SmartPrep2 System	Significant clinical improvement; Good stability of implant and complete filling in 80% on MRI; Hyaline-like cartilage	N/A
Gobbi et al. [57]	Cartilage/2015	Prospective comparative study/37	M-ACI (43.1) vs. BMAC (45.4)	≥36.0	Mean size 7.1 cm^2^ (M-ACI) vs. 5.5 cm^2^ (BMAC) ICRS grade IV lesion/patella or trochlea	M-ACI (19) vs. BMAC (18)	HA scaffold + fibrin glue	60 mL/SmartPrep2 System	Significant clinical improvement in both groups; no significant difference between the groups; Complete filling on MRI 76.0% (M-ACI) vs. 81.0% (BMAC); Hyaline-like features	N/A
Gobbi et al. [68]	Am J Sports Med/2019	Case series/23	48.5	96.0	Mean 6.5 cm^2^ sized ICRS grade IV lesion /MFC or patellar or trochlea	BMAC + HA-based scaffold	HTO; TTO; ACLR; LR	60 mL/SmartPrep2 System	Good to excellent long-term clinical outcomes in full-thickness cartilage injury of the knee joint	N/A
Enea et al. [56]	Knee/2015	Case series/9	43.0	29.0	Mean size 2.6 cm^2^ with chondral defect Outerbridge type III- IV/MFC or LFC	BMAC + fibrin glue	Collagen membrane; Microfx.or partial menicectomy or synovectomy	60 mL/MarrowStim Concentration Kit	Significant clinical improvement; Almost normal arthroscopic appearance of repaired cartilage; Regeneration potential to hyaline-like cartilage	N/A

BMAC, bone marrow aspirate concentrate; ICRS, International Cartilage Regeneration and Joint Preservation Society; MFC, medial femoral condyle; LFC, lateral femoral condyle; HA, hyaluronic acid; MRI, magnetic resonance image; N/A, non-available; Microfx., microfracture; vs., versus; MSCs, mesenchymal stem cells; M-ACI, matrix-induced autologous chondrocyte implantation; HTO, high tibial osteotomy; TTO, tibial tubercle osteotomy; ACLR, anterior cruciate ligament reconstruction; LR, lateral release.

**Table 3 ijms-21-03224-t003:** Details of the clinical trials of BMAC focusing on osteoarthritis.

Authors	Publications/Year	Study Design/No. of Patients	Mean Age (Year)	Mean Follow-Up Period (Months)	OA Grades	Treatment	Additional Factors	Harvest Volume/Kit Used	Outcomes	Complications
Hauser et al. [70]	Clin Med Insights Arthritis Musculoskelet Disord/2013	Case series/7 (hip, knee, ankle OA)	64.0	7.1	N/A	Whole bone marrow injection	Dextrose prolotherapy	Not concentrated	Substantial gain in pain relief & functionality	N/A
Centeno et al. [69]	Biomed Res Int/2014	Retrospective comparative study/840	54.3 vs. 59.9	10.4 vs. 10.7	K–L grade 1,2,3,4	BMAC alone (616) vs. BMAC + adipose graft (224)	PRP	Manual aspiration in a sterile ISO-7 class clean room and in ISO-5 class laminar flow cabinets	Encouraging clinical outcomes with a low rate of AEs; Better results in K–L 2 than K–L 3-4 (2.2 times); Adipose graft did not provide additional benefit	AEs rates 6.0% (BMAC alone) vs. 8,9% (BMAC + adipose graft)
Shapiro et al. [72]	Am J Sports Med/2017	Prospective RCT/25 (bilateral knee OA)	60.0	6.0	K–L grade 1,2,3	BMAC vs. Saline	PRP	52 mL/Automated centrifuge (Magellan Autologous Platelet Separator System)	Pain relief did not differ significantly between both knees	N/A
Kim JD et al. [59]	Eur J Orthop Surg Traumatol/2014	Case series/75	60.7	8.7	K–L grade 1,2,3,4	BMAC	Arthroscopic debridement; Microfx.; HTO	120 mL/SmartPrep2 System	Significant clinical improvement; Better results in K–L 1-3 than K–L 4	Swelling: 92.0% Pain: 41.3%

BMAC, bone marrow aspirate concentrate; OA, osteoarthritis; N/A, non-available; vs., versus; K–L grade, Kellgren–Lawrence grade; PRP, platelet rich plasma; AEs, adverse events; RCT, randomized controlled trial; Microfx., microfracture; HTO, high tibial osteotomy.

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
