# Peer review of "Bone Marrow Aspirate Concentrate: Its Uses in Osteoarthritis"

_ijms, 2020, doi:10.3390/ijms21093224_

Round 1
Reviewer 1 Report
Although this paper is nicely written, this topic has already been published in several other journals. The authors provided some of the research for bone marrow aspirate concentration in the treatment of us arthritis but several clinical studies are missing. In addition some information regarding the constituents of bone marrow are not included in this article. Therefore, do not feel that this manuscript adds to the current literature.
I did not feel that this paper adds much to what has ALREADY been published.
Author Response
First of all, we are deeply grateful for your kind and excellent comments on the manuscript.
Reviewer 1’s comments: Although this paper is nicely written, this topic has already been published in several other journals. The authors provided some of the research for bone marrow aspirate concentration in the treatment of us arthritis but several clinical studies are missing. In addition, some information regarding the constituents of bone marrow are not included in this article. Therefore, do not feel that this manuscript adds to the current literature.
I did not feel that this paper adds much to what has ALREADY been published.
Response: Thank you for your thoughtful review of our article. Unfortunately, we are sorry to disagree with your opinion. As you pointed out, some studies of BMAC have been reported. However, as in this review article, few studies have systematically analyzed the entire contents of BMAC, from the components of BMAC to the latest related technologies. In addition, contrary to your point of view, this study covers various constituents of BMAC, including cellular components, mesenchymal stem cells, growth factors, and cytokine. Although its contents had been already reported, we believe that this review article is worth publishing as an article of IJMS. Please, look forward to considering the positive aspects of our article.

Reviewer 2 Report
Reviewer Comments to author
Bone Marrow Aspirate Concentrate: Its Uses in Osteoarthritis
GENERAL COMMENTS
Interesting narrative review
Normal length.
Good english
ABSTRACT
It is concise, and the content is correct
TITLE
The title is correct.
INTRODUCTION
Provide the necessary background for the review
METHODS and MATERIAL
Do not have
RESULTS
Do not have
CONTENT
The manuscript's authors comment on basic science and clinical studies of BMAC and Osteoarthritis
Future trends
CONCLUSION
Do not have
REFERENCES
Please add the latest studies Gobbi 2018/19
FIGURES
Please add BMAC pictures
Author Response
First of all, we are deeply grateful for your kind and excellent comments on the manuscript.
Reviewer 2’s comments:
GENERAL COMMENTS
Interesting narrative review
Normal length.
Good English.
Response: Thank you for your positive comments
ABSTRACT
It is concise, and the content is correct.
Response: Thank you for your positive comments
TITLE
The title is correct.
Response: Thank you for your positive comments
INTRODUCTION
Provide the necessary background for the review
Response: Following your comment, we have added the sentence regarding the necessary background for this review article. Thank you.
Line 45-47: Although several studies on BMAC have been introduced, there has been paucity in the systemically organized review on its overall contents.
METHODS and MATERIAL
Do not have
Response: No need to answer.
RESULTS
Do not have
Response: No need to answer.
CONTENT
The manuscript's authors comment on basic science and clinical studies of BMAC and Osteoarthritis
Future trends
Response: Thank you for your positive comments
CONCLUSION
Do not have
Response: No need to answer.
REFERENCES
Please add the latest studies Gobbi 2018/19
Response: Following your comment, we have added the recent article of Gobbi et al [Am J Sports Med, 2018]. Moreover, we have added the reference in the section 7 and table 2. As a result, we have also modified the order of the references. Thank you.
Line 221-223: Recently, they reported excellent long-term clinical outcomes of hyaluronic acid-based scaffold embedded with BMAC in a full-thickness cartilage defect [68].
Table 2. Details of the clinical trials of BMAC focusing on cartilage defect.
Authors |
Publications / Year |
Study design / No. of patients |
Mean age (year) |
Mean follow-up period (months) |
Defect / Location |
Treatment |
Additional factors |
Harvest volume/ kit used |
Outcomes |
Complications |
Gobbi et al. [68] |
Am J Sports Med / 2019 |
Case series / 23 |
48.5 |
96.0 |
Mean 6.5 cm2 sized ICRS grade IV lesion / MFC or patellar or trochlea |
BMAC + HA-based scaffold |
HTO; TTO; ACLR; LR |
60 ml/ SmartPrep2 System |
Good to excellent long-term clinical outcomes in full-thickness cartilage injury of the knee joint |
N/A |
FIGURES
Please add BMAC pictures
Response: Following your comment, we have added the picture related to BMAC preparation and injection, as a Figure 2, with figure legend (Line 191-193).
Line 221-223: Figure 2. BMAC preparation and knee joint injection. (A) bone marrow aspiration at anterior iliac rim. (B) After centrifugation and some procedure, dark-colored BMAC (white arrow) was obtained. (C) BMAC injection to the knee joint with osteoarthritis.
Thank you.

Round 2
Reviewer 1 Report
This paper is very well done and complete. It should be very helpful to both scientists and clinicians.
Author Response
Dear reviewer
We are deeply grateful for your excellent comments on our article. We are very honored to be accepted by IJMS. Finally, we have checked the last minor issues of our article for complete acceptance. Two authors were added during the Revision process, we have changed it, and also submitted an authorship change form. Thank you.
Sincerely yours,
Authors
This manuscript is a resubmission of an earlier submission. The following is a list of the peer review reports and author responses from that submission.
Round 1
Reviewer 1 Report
Although this paper is nicely written, this topic has already been published in several other journals. The authors provided some of the research for bone marrow aspirate concentration in the treatment of us arthritis but several clinical studies are missing. In addition some information regarding the constituents of bone marrow are not included in this article. Therefore, do not feel that this manuscript adds to the current literature.
Reviewer 2 Report
I reviewed the manuscript of Gi Beom Kim and Gun Woo Lee entitled "Bone Marrow Aspirate Concentrate: Its Uses in Osteoarthritis".
The manuscript provides a general overview of the properties of the Bone Marrow Mesenchymal Stem Cells aspirate and its use for the treatment of osteoarthritis.
The manuscript is clear and the topics are well organized through the review. It is well written, and only minor revisions are requested to be suitable for the publication. In particular:
More references should be provided to support some claims:
- rows 139,140: please add more references describing the immunomodulatory and anti-inflammatory activities of MSC
- rows 172, 173: please change the current references with more appropriate and significant references (ref [60]- results on 4 patients are not significant; [61] too old for the claim, being published on 2011)
- In the paragraph "Application modalities", the possibility to perform injections into the subchondral bone should be mentioned, because it has been suggested to be beneficial for the treatment of knee OA because of the current evidence on the role of the subchondral bone in the pathogenesis of the knee OA.